# The Point Where Reality Meets Fantasy: Mixed Adversarial Generators for Image Splice Detection

**Vladimir V. Kniaz**[1,2]**, Vladimir A. Knyaz**[1,2]
[1]State Res. Institute of Aviation Systems (GosNIIAS)
125319, 7, Victorenko str., Moscow, Russia
{vl.kniaz, knyaz}@gosniias.ru
[2]Moscow Institute of Physics and Technology (MIPT)
141701, 9 Institutskiy per., Dolgoprudny, Russia

**Fabio Remondino**
Fondazione Bruno Kessler (FBK)
Via Sommarive 18, Trento, Italy
remondino@fbk.eu

## Abstract

Modern photo editing tools allow creating realistic manipulated images easily. While fake images can be quickly generated, learning models for their detection is challenging due to the high variety of tampering artifacts and the lack of large labeled datasets of manipulated images. In this paper, we propose a new framework for training of discriminative segmentation model via an adversarial process. We simultaneously train four models: a generative retouching model $G_R$ that translates manipulated image to the real image domain, a generative annotation model $G_A$ that estimates the pixel-wise probability of image patch being either real or fake, and two discriminators $D_R$ and $D_A$ that qualify the output of $G_R$ and $G_A$. The aim of model $G_R$ is to maximize the probability of model $G_A$ making a mistake. Our method extends the generative adversarial networks framework with two main contributions: (1) training of a generative model $G_R$ against a deep semantic segmentation network $G_A$ that learns rich scene semantics for manipulated region detection, (2) proposing per class semantic loss that facilitates semantically consistent image retouching by the $G_R$. We collected large-scale manipulated image dataset to train our model. The dataset includes 16k real and fake images with pixel-level annotations of manipulated areas. The dataset also provides ground truth pixel-level object annotations. We validate our approach on several modern manipulated image datasets, where quantitative results and ablations demonstrate that our method achieves and surpasses the state-of-the-art in manipulated image detection. We made our code and dataset publicly available [1].

## 1 Introduction

While every image captured by the human eye is real, digital photos can be easily manipulated to present scenes that never existed in reality. Such manipulated image can be easily generated by copying the part of one image into another. This image manipulation is called an image splice and can be used maliciously to create fake news or change historical photos [1]. Recent research [1, 2] suggests that training a model for splice localization is more challenging than other types of object detection problems as the domain of manipulated images is extensive and diverse. Therefore, the collection of the representative training dataset is difficult. Moreover, the forger can adapt to the detection algorithm by changing the manipulation technique. This principle is used in Generative Adversarial Networks (GANs) to train a generator network to synthesize images from noise [3], text descriptions [4], scene graphs [5] or by image-to-image translation [6, 7, 8, 9]. Fake images produced by the generator are evaluated against real images by an adversarial discriminator network that learns

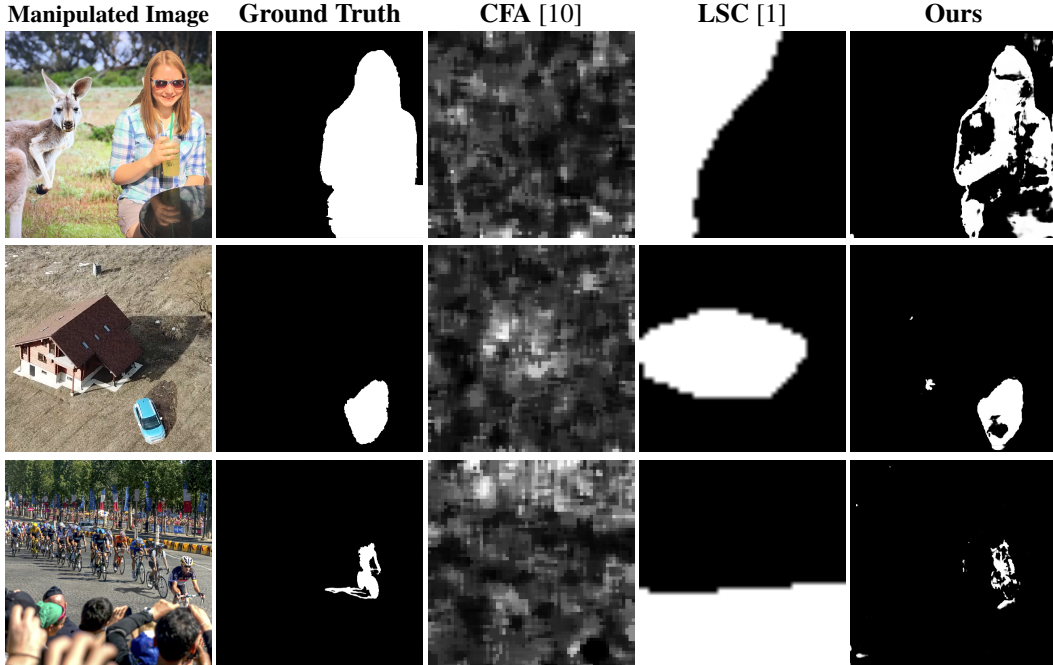

| Manipulated Image | Ground Truth | CFA [10] | LSC [1] | Ours |

Figure 1: Comparison against two state-of-the-art methods on our *FantasticReality* dataset (Section 3.3). Our results are shown in the last column. Zoom in for details.

to classify them as 'real' or 'fake.' Even though no 'fake' images exist in the training dataset, the discriminator successfully learns to detect them during the training process. We hypothesize that adversarial training of an image-to-image translation generator against a splice localization generator can improve the splice localization accuracy.

In this paper, we propose a Mixed Adversarial Generators (MAG) framework, in which we simultaneously train four models: a generative retoucher $G_R$, an adversarial generative annotator $G_A$, and two discriminators $D_R$ and $D_A$ that qualify the output of $G_R$ and $G_A$. The aim of our retoucher $G_R$ is suppressing image tampering artifacts in the input image splices from the training dataset. We train our adversarial annotator $G_A$ to predict splice localization masks in the 'retouched' images generated by the retoucher $G_R$. The adversarial loss provided by the annotator $G_A$ forces the retoucher $G_R$ to mask those particular tampering artifacts that allow $G_A$ to detect the image splice. Unlike other splice detection models, our annotator $G_A$ learns to adapt to changing tampering techniques of the retoucher $G_R$. Therefore, our annotator $G_A$ receives a new sample from the manipulated image domain every iteration. Moreover, with the increasing epoch samples are becoming more complex. To further increase the splice localization rate, we train our annotator $G_A$ to predict object classes for the input image. Resulting semantic labeling is used to provide a semantic consistency loss for the retoucher $G_R$. The semantic loss forces the output of the retoucher $G_R$ to present objects of the same semantic classes as the input image.

Our adversarial generators extend the GAN framework with two key contributions: (1) training of a generative model $G_R$ against a deep semantic segmentation network $G_A$ that learns rich scene semantics for manipulated region detection, (2) proposing per class semantic loss that facilitates semantically consistent image retouching by the $G_R$. Unlike the recently proposed Sem-GAN model [11], we do not use the pertained segmentation model but train it adversarially. We perform a comprehensive evaluation of our MAG framework, where quantitative results and ablations demonstrate that our annotator $G_A$ achieves and surpasses the state-of-the-art in splice localization on several challenging image splice datasets (see Figure 1 and 3).

We evaluate our retoucher $G_R$ on image-to-image translation tasks to demonstrate that our MAG framework is not limited to the splice localization task. Semantic loss function allows us to train challenging image-to-image translation tasks that are unfeasible for baselines. We also introduce a new *FantasticReality* dataset that includes 16k image splices with pixel-level ground truth annotations

of manipulated areas, and instance and class labels for ten object categories. We made our code and the dataset publicly available.

## 2  Related Work

**Splice detection.** Modern splice detection methods fall into three categories: tampering artifacts-based approaches leverage local discrepancies in image noise [12, 13, 14, 15, 16, 17, 18], compression artifacts [19, 20, 21, 22], or camera's color filter array inconsistencies [10, 23, 24, 25, 26, 27, 28, 29, 30, 31] to detect tampered image regions; consistency-based methods [32, 1] compare pairs of local image patches to localize image areas, where predicted camera model [33, 32] or image metadata [1] are inconsistent with the rest of the image; deep learning-based methods [34, 35, 1, 2, 36] detect image splice regions either by comparison of local patches in a siamese network [1] or using fully convolutional networks [2] to predict labeling of the tampered regions. While many digital image forensic datasets were introduced recently [37, 38, 39, 40, 41], they usually include only several hundreds of photos and do not provide enough of training data for modern methods. Related to our multi-task annotation prediction, Salloum et al. [2] have proposed multi-task training to localize tampered regions and their edges.

**Image-to-image translation.** Modern methods for image generation conditioned by an input image are trained in either supervised [6, 42, 43, 11, 44], unsupervised [7, 8, 9, 45, 46, 47] or mixed [48] setting. Unsupervised approaches are trained on an unpaired dataset leveraging the latent space assumption [8], the cycle consistency loss [7] or other criteria to learn a mapping from source to target domain. Recent research demonstrates exciting progress in multimodal image-to-image translation [49, 42]. Related to our semantic consistency loss function are the loss functions proposed in `Sem-GAN` [11] and `InstaGAN` [50] models. Unlike our `MAG` framework `Sem-GAN` model leverages a pretrained segmentation model to provide semantic loss. Unlike `InstaGAN` [50] model, our retoucher generator $G_R$ does not require instance masks as an input. Closely related to our retoucher generator $G_R$, Mejjati et al. [9] propose to use attention guided training to perform translation only for the target object.

Most of the modern approaches in the image-to-image translation are based on Generative Adversarial Networks [3], which can capture the sample distribution in the target domain using an adversarial game of two players. Recent research demonstrates that GANs can solve more challenging tasks than image-to-image translation. They can learn complex transforms between physically different domains such as image-to-thermal translation [51, 52, 53, 54, 55], image-to-voxel model transformation [56, 57], and image synthesis from audio data [58]. In our `MAG` framework, we replace the discriminator network with an adversarial annotator generator $G_A$. While the discriminator predicts a scalar probability of an input image being either real or fake, our annotator generator $G_A$ predicts a pixel-level probability map of an image patch being either authentic image or splice.

## 3  Mixed Adversarial Generators

Our goal is training two generator networks adversarially: a splice retoucher $G_R$ and a splice localization annotator $G_A$. We consider three domains: the input domain $\mathcal{A} \in \mathbb{R}^{W \times H \times 3}$ of potentially manipulated images, the authentic domain $\mathcal{B} \in \mathbb{R}^{W \times H \times 3}$ of untampered images, and the output domain $\mathcal{C} \in [0,1]^{W \times H \times (2+K)}$ of splice localization and class segmentation masks, where $K$ is the number of predicted object classes. While an image $\boldsymbol{A} \in \mathcal{A}$ may be either authentic or tampered, all images $\boldsymbol{B} \in \mathcal{B}$ are authentic, $\mathcal{B} \subset \mathcal{A}$. We use assumptions made by Salloum et al. [2] as the starting point for our generator $G_A$. Specifically, we train our generator $G_A$ for multi-task prediction of splice segmentation mask $\boldsymbol{C}_m \in [0,1]^{W \times H}$, splice edge mask $\boldsymbol{C}_e \in [0,1]^{W \times H}$, and object class segmentation $\boldsymbol{C}_s \in [0,1]^{W \times H \times K}$. Therefore, we learn a mapping $G_A : (\boldsymbol{A}) \rightarrow \boldsymbol{C}$, where $\boldsymbol{A} \in \mathcal{A}$ is an input potentially manipulated image, $\boldsymbol{C} \in \mathcal{C}$ is an output tensor obtained by concatenation of $\boldsymbol{C}_m, \boldsymbol{C}_e, \boldsymbol{C}_s$. The goal of our retoucher generator $G_R$ is learning a mapping from manipulated image domain $\mathcal{A}$ to the authentic domain $\mathcal{B}$. To this end, the aim of adversarial training of the $G_R$ is maximizing the probability of an annotator $G_A$ making a mistake in splice detection of the retouched image $\hat{\boldsymbol{B}}$. We believe that the retoucher $G_R$ in the training loop facilitates our annotator $G_A$ to learn complicated splice retouching approaches. We use attention-guided learning assumption made by Mejjati et al. [9] as the starting point for our retoucher $G_R$. We observe the similarity between the attention map proposed by Mejjati et al. [9] and the alpha channel used for the splice

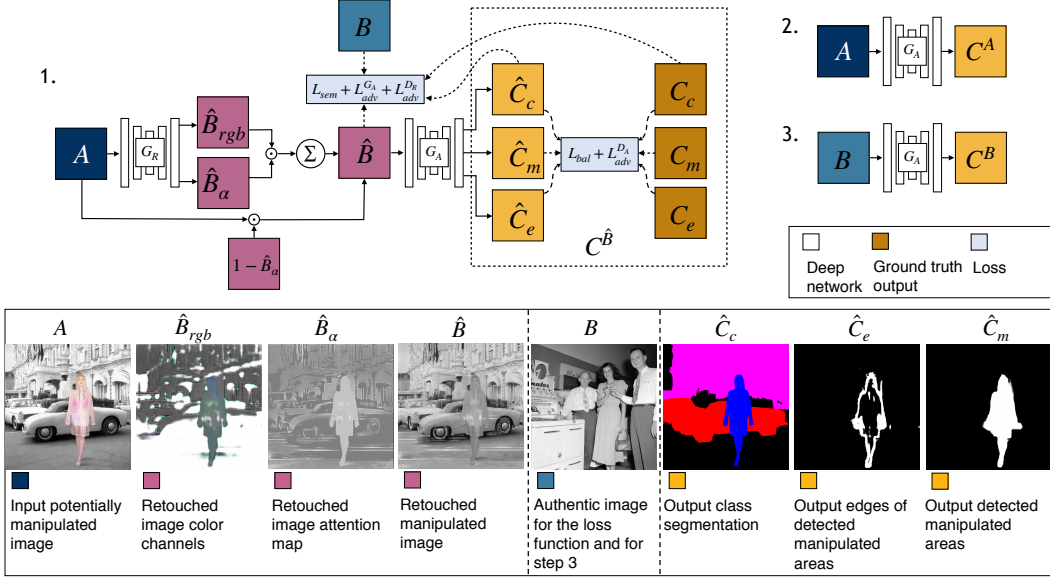

Figure 2: **Our proposed pipeline**: We want our annotator $G_A$ to predict annotations correctly for three kinds of images: retouched spliced images $\hat{\boldsymbol{B}} = G_R(\boldsymbol{A})$ (1), original spliced images $\boldsymbol{A} \in \mathcal{A}$ from the training dataset (2), and authentic images $\boldsymbol{B} \in \mathcal{B}$ (3). Our retoucher $G_R$ learns to hide a wide range of tampering artifacts such as modern-to-retro photo translation, blurring of tampering edges, and compensating light source inconsistencies. During the training, we feed manipulated images $\boldsymbol{A}$, retouched images $\hat{\boldsymbol{B}}$, and authentic images $\boldsymbol{B}$ to our splice localization annotator $G_A$.

generation. We hypothesize that attention-guided learning of our retoucher $G_R$ allows us to model splice generation with layers in photo-editing applications, e.g., GIMP or Photoshop. We learn a mapping $G_R : (\boldsymbol{A}) \rightarrow (\hat{\boldsymbol{B}}_{rgb}, \hat{\boldsymbol{B}}_\alpha)$, where $\hat{\boldsymbol{B}}_{rgb} \in \mathbb{R}^{W \times H \times 3}$ is an image with the retouched splice area, and $\hat{\boldsymbol{B}}_\alpha \in [0,1]^{W \times H}$ is the attention map. We obtain the target retouched splice image $\hat{\boldsymbol{B}}$ similarly to [9] by

$$\hat{\boldsymbol{B}} = \hat{\boldsymbol{B}}_\alpha \odot \hat{\boldsymbol{B}}_{rgb} + (1 - \hat{\boldsymbol{B}}_\alpha) \odot \boldsymbol{A}, \tag{1}$$

where $\odot$ is an element-wise product. Our proposed pipeline is presented in Figure 2. We train two discriminator networks $D_R$, $D_A$ to provide adversarial losses for the output of our generators $G_R$ and $G_A$. The architecture and the loss function of the retoucher $G_R$ are presented in Section 3.1, whereas the structured loss function of the annotator $G_A$ is described in Section 3.2.

## 3.1 Retoucher Generator $G_R$

**Architecture.** We use the U-Net generator architecture [59] as the starting point for our retoucher $G_R$. While skip connections of the U-Net generator facilitate robust learning of tampering techniques by our retoucher $G_R$, deconvolutional layers often introduce checkerboard artifacts in output images. Our annotator $G_A$ quickly learns checkerboard features to detect images produced by our retoucher $G_R$. To avoid such a scenario, we replaced deconvolutional layers with an upsample layer followed by a convolutional layer, inspired by the architecture proposed in [60]. We term the resulting architecture that is free from the checkerboard artifacts as U-Net-UC (see supplementary material Table 1).

**Loss function.** Three loss functions govern the training process for our retoucher $G_R$: $\mathcal{L}_{sem}^{G_A}$, $\mathcal{L}_{adv}^{G_A}$, and $\mathcal{L}_{adv}^{D_R}$, where a superscript indicates the network providing the loss. The aim of our semantic

consistency loss function $\mathcal{L}_{sem}^{G_A}$ is to make the classes of objects in the output image $\hat{B}$ recognizable by our annotator $G_A$

$$\mathcal{L}_{sem}^{G_A}(C_s, \hat{C}_s) = \mathbb{E}_{B \sim p(B)}\left[\left|\left|C_s - \hat{C}_s\right|\right|_1\right], \tag{2}$$

where $\hat{C}_s = G_A(\hat{B})_s$ is the class segmentation produced by our annotator $G_A$, $C_s$ is the ground truth class segmentation. Our adversarial annotator loss $\mathcal{L}_{adv}^{G_A}$ stimulates our retoucher $G_R$ to mask tampering artifacts in the input sliced images. In other words, we want to maximize the probability of our annotator $G_A$ making a mistake in splice localization $\hat{C}_m = G_A(\hat{B})_m$

$$\mathcal{L}_{adv}^{G_A}(C_m, \hat{C}_m) = \mathbb{E}_{B \sim p(B)}\left[\left|\left|\mathbf{0}_{W,H} - \hat{C}_m\right|\right|_1\right], \tag{3}$$

where $\mathbf{0}_{W,H}$ is the a splice localization filled with zeros. Finally, we use a discriminator's $D_R$ adversarial loss function to make our image realistic globally

$$\mathcal{L}_{adv}^{D_R}(\hat{B}) = \mathbb{E}_{B \sim p(B)}\left[\log(1 - D_R(\hat{B}))\right]. \tag{4}$$

We obtain the final energy to be optimized by combining all losses

$$\mathcal{L}_R(C_s, \hat{C}_s, C_m, \hat{C}_m, \hat{B}) = \lambda_{sem}^{G_A} \cdot \mathcal{L}_{sem}^{G_A} + \lambda_{adv}^{G_A} \cdot \mathcal{L}_{adv}^{G_A} + \lambda_{adv}^{D_R} \cdot \mathcal{L}_{adv}^{D_R}, \tag{5}$$

where we use the loss hyper-parameters $\lambda_{sem}^{G_A} = 10$, $\lambda_{adv}^{G_A} = 10$, $\lambda_{adv}^{D_R} = 0.25$ in our experiments.

## 3.2 Annotator Generator $G_A$

**Loss function.** We train our annotator $G_A$ utilizing a combination of our balanced $L^1$ loss function $\mathcal{L}_{bal}$ and an adversarial loss $\mathcal{L}_{adv}^{D_A}$. We observe that training our annotator $G_A$ using the $L^1$ distance $||C - \hat{C}||$ between the ground-truth and predicted annotations results in a large number of false negatives in splice localizations. We hypothesize that making the penalty for false negatives and false positives equal for each image can improve the overall splice localization score. We implement this hypothesis in our balanced loss function based on the Dice loss [61]

$$\mathcal{L}_{bal}(C, \hat{C}) = \underbrace{\sum_{i=1}^{2+K} \frac{|C_i \cap (1 - \hat{C}_i)|}{|C_i|}}_{\text{False negatives}} + \underbrace{\sum_{i=1}^{2+K} \frac{|(1 - C_i) \cap \hat{C}_i|}{|1 - C_i|}}_{\text{False positives}}, \tag{6}$$

where $i$ is the index of an annotation channel. Channel $C_1$ provides a splice mask annotation $C_m$, channel $C_2$ provides a splice edges annotation $C_e$. The predicted class labels are given by $C_i$ for $i \in \{3, 4, \ldots, 2 + K\}$, where $K$ is the number of classes ($K = 10$ in our experiments). The area of predicted annotations (white area) in the channel $C_i$ is given by $|C_i|$, the background area (black area) in the channel $C_i$ is given by $|1 - C_i|$.

We want our annotator $G_A$ to predict annotations correctly for three kinds of images: original spliced images $A \in \mathcal{A}$ from the training dataset, retouched spliced images $\hat{B} = G_R(A)$, and authentic images $B \in \mathcal{B}$. Therefore, for each iteration, we evaluate the loss $\mathcal{L}_{bal}$ on three pairs of ground truth and predicted annotations: $(C^{A'}, \hat{C}^A)$, $(C^{A'}, \hat{C}^{\hat{B}})$, $(C^B, \hat{C}^B)$. We use the superscript to denote the corresponding color image for the annotation. Please, note that both original spliced image $A$ and the retouched spliced image $\hat{B}$ have the same annotation $C^{A'}$ with an adversarial class segmentation mask $C_s^{A'} = C_s^A \odot (1 - C_m^A)$. We want to train our annotator $G_A$ to predict class segmentation adversarially: it must generate the correct class annotations only for authentic image areas and predict empty class annotations for manipulated regions. Specifically, we multiply our ground truth semantic segmentation $C_s^A$ by an inverted splice localization mask $(1 - C_m^A)$. The

multiplication by the inverted mask leaves authentic areas untouched and removes annotations for manipulated regions.

The aim of our adversarial loss $\mathcal{L}_{adv}^{D_A}$ is to avoid blurry output splice localization masks [6]. It is provided by a conditional discriminator $D_A$ with PatchGAN architecture [6]

$$\mathcal{L}_{adv}^{D_A}(\hat{\boldsymbol{B}}, \hat{\boldsymbol{C}}^{\hat{\boldsymbol{B}}}) = \mathbb{E}_{\boldsymbol{B} \sim p(\boldsymbol{B})}\Big[ \log(1 - D_A(\hat{\boldsymbol{B}}, \hat{\boldsymbol{C}}^{\hat{\boldsymbol{B}}})) \Big]. \tag{7}$$

We obtain the resulting energy to optimize by combining four loss functions

$$\mathcal{L}_A = \lambda_{bal}\Big( \mathcal{L}_{bal}(\boldsymbol{C}^{\boldsymbol{A}'}, \hat{\boldsymbol{C}}^{\boldsymbol{A}}) + \mathcal{L}_{bal}(\boldsymbol{C}^{\boldsymbol{A}'}, \hat{\boldsymbol{C}}^{\hat{\boldsymbol{B}}}) + \mathcal{L}_{bal}(\boldsymbol{C}^{\boldsymbol{B}}, \hat{\boldsymbol{C}}^{\boldsymbol{B}}) \Big) + \lambda_{adv}^{D_A} \cdot \mathcal{L}_{adv}^{D_A}(\hat{\boldsymbol{B}}, \hat{\boldsymbol{C}}^{\hat{\boldsymbol{B}}}), \tag{8}$$

where we use the loss hyper-parameters $\lambda_{bal} = 1, \lambda_{adv}^{D_A} = 1$ in our experiments.

### 3.3 FantasticReality Dataset

We collected large-scale image tampering dataset with 16k authentic and 16k tampered images to perform extensive training and evaluation of our MAG model. Compared to previous datasets [37, 38, 39, 40], our *FantasticReality* dataset is more extensive in terms of scene variety and image count. To the best of our knowledge, it is the first tampering dataset that provides both tampering masks and instance and class labels for each image. For each authentic and tampered image, we manually generated instance and class segmentation for ten object classes: person, car, truck, van, bus, building, cat, dog, tram, boat. Examples from the dataset are presented in Figure 1 in the supplementary material.

## 4 Experiments

We perform extensive experiments to evaluate our MAG model on splice localization. We compare our model to three modern state-of-the-art deep learning splice detection frameworks: ManTra [62], LSC [1], MFCN [2]. We provide a comparison to non-deep learning methods to be consistent with LSC: NOI [18], CFA [10], DCT [19]. ManTra-Net (ManTra) [62] is a self-supervised model that learns to classify 385 image manipulation types. Learned Self-Consistency (LSC) [1] is a self-supervised model. Multi-Task Fully Convolutional Network (MFCN) [2] leverages a deep two-stream architecture to predict splice mask and splice edge mask. Noise Variance (NOI) [18] leverages wavelet analysis to detect inconsistency in noise patterns. Color Filter Array (CFA) [10] searches for inconsistencies in artifacts of demosaicking algorithm to detect tampered regions. JPEG DCT [19] leverages inconsistencies of JPEG blocking artifacts to detect tampered image regions. For the LSC algorithm, we use a pertained model provided by authors. We implemented the MFCN model and train it on the training split of our FantasticReality dataset. We train our MAG model on the 'Rough' split of our FantasticReality dataset. We use a batch size of one and an Adam solver with initial learning rate of $2 \cdot 10^{-4}$. We trained our MAG model for 400 epochs.

We perform evaluation on five manipulated image datasets *CASIA v2.0* [37], *Carvalho* [38], *Columbia* [39], *Realistic Tampering* [40] and our *FantasticReality* dataset. For the fair evaluation, we downscale all images to match the input size $512 \times 512$ of our annotator generator $G_R$. We use the downscaled images to evaluate all baselines and our framework. If two images are used for splice generation, the choice of 'authentic' and 'tampered' regions is ambiguous. To avoid ambiguity, we follow the method proposed in [1]. Namely, we compare the areas of the 'background' image and the 'pasted' images. We define the smaller region as the tampered. If the regions are equal, we calculate the mAP score for the original tampering mask and an inverted mask. We use the higher score and term it permuted mAP (p-mAP) similar to [1]. For additional details on the evaluation protocol, please, refer to the supplementary material. Furthermore, we perform ablation studies to demonstrate the influence of each component of our framework on the resulting performance.

### 4.1 Annotator Generator $G_A$ Evaluation

**Splice Localization.** We evaluate our model and baselines on the task of splice localization using ground-truth masks of spliced regions. Specifically, we want our model to predict a per-pixel

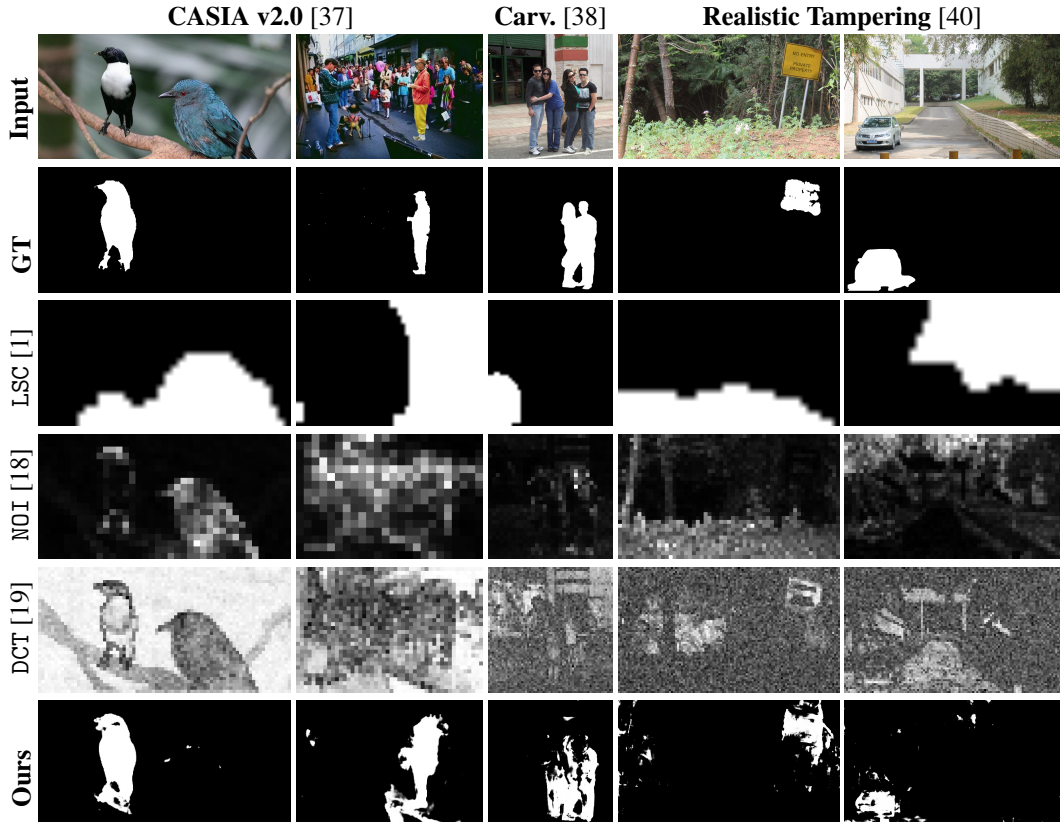

Figure 3: Comparison against the state-of-the-art methods on image splices from CASIAv2, Carvalho and Realistic Tampering datasets. Our results are shown in the last row. Zoom in for details.

| Dataset | CASIA v2.0 [37] | | | Columbia [39] | | | RT [40] | | | Carvalho [38] | | | FantasticReality | | |
|---|---|---|---|---|---|---|---|---|---|---|---|---|---|---|---|
| Metric | mAP | p-mAP | cIOU | mAP | p-mAP | cIOU | mAP | p-mAP | cIOU | mAP | p-mAP | cIOU | mAP | p-mAP | cIOU |
| LSC | 0.32 | 0.41 | 0.47 | 0.25 | 0.44 | 0.41 | 0.33 | 0.47 | 0.52 | 0.15 | 0.33 | 0.24 | 0.17 | 0.45 | 0.36 |
| CFA | 0.37 | 0.40 | 0.42 | 0.39 | 0.44 | 0.44 | 0.45 | 0.48 | 0.49 | 0.32 | 0.32 | 0.33 | 0.45 | 0.50 | 0.48 |
| NOI | 0.29 | 0.45 | 0.46 | 0.26 | 0.43 | 0.40 | 0.48 | 0.45 | 0.50 | 0.21 | 0.31 | 0.21 | 0.18 | 0.49 | 0.29 |
| LSC | 0.35 | 0.39 | 0.35 | 0.22 | 0.42 | 0.43 | 0.37 | 0.49 | 0.48 | 0.16 | 0.37 | 0.25 | 0.26 | 0.51 | 0.41 |
| MFCN | 0.36 | 0.41 | 0.48 | 0.27 | 0.45 | 0.42 | 0.41 | 0.51 | 0.36 | 0.42 | 0.36 | 0.37 | 0.40 | 0.51 | 0.46 |
| ManTra | 0.40 | 0.40 | 0.45 | 0.48 | 0.48 | 0.58 | **0.50** | 0.50 | 0.54 | 0.33 | 0.33 | 0.38 | 0.57 | 0.57 | 0.73 |
| No $G_R$ | 0.41 | 0.35 | 0.12 | 0.34 | 0.32 | 0.29 | 0.28 | 0.35 | 0.18 | 0.20 | 0.37 | 0.29 | 0.27 | 0.45 | 0.36 |
| Single-task | 0.12 | 0.15 | 0.24 | 0.11 | 0.19 | 0.21 | 0.17 | 0.16 | 0.14 | 0.19 | 0.15 | 0.18 | 0.12 | 0.17 | 0.21 |
| Ours | **0.74** | **0.74** | **0.76** | **0.69** | **0.69** | **0.77** | 0.50 | **0.51** | **0.55** | **0.48** | **0.48** | **0.56** | **0.61** | **0.61** | **0.76** |

Table 1: **Splice Localization:** We evaluate our model on 5 datasets using mean average precision (mAP, permuted-mAP) over pixels and per class IOU (cIOU).

probability of an image patch being tampered. We present results in terms of mAP, permuted mAP [1], and per class Intersection over Union (cIOU) in Table 1 and in Figure 3. Our MAG model achieves state-of-the-art in splice localization on all datasets. The LSC model fails to detect splices when authentic and spliced regions originate from the same camera model and share similar camera metadata.

**Ablation Study.** We evaluate the necessity of all components of our MAG framework by comparing the splice localization accuracy of several ablated versions of our model presented in Table 1 and Figure 4. Firstly, we evaluate the performance of annotator $G_A$ trained without retoucher $G_R$ (No $G_R$). Both qualitative and quantitative results demonstrate that the competition of two generators is the critical

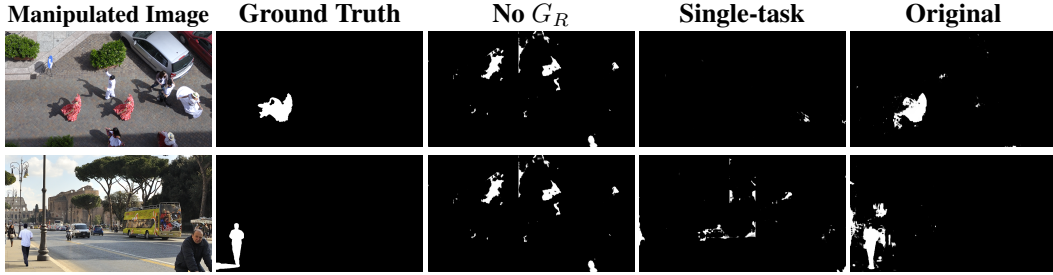

| Manipulated Image | Ground Truth | No $G_R$ | Single-task | Original |
|---|---|---|---|---|

Figure 4: Qualitative results for ablated versions of our MAG framework evaluated on Realistic Tampering dataset.

component of our MAG framework. Secondly, we evaluate our framework trained for the single task of predicting splice area annotations. The results prove that multi-task training outperforms the single-task version of our model (Single-task).

## 4.2 Semantic-guided Retoucher Generator $G_R$ Evaluation

Examples in Figures 5 and 6 demonstrate how our retoucher $G_R$ gradually removes the tampering artifacts in the input splice $A$ with an increasing epoch. While other deep learning splice detection methods receive both realistic and rough splices from the first training epoch, our annotator $G_A$ sees only rough splices at the first epoch. With an increasing epoch, retoucher $G_R$ produces more complicated splices, which allows $G_A$ to focus attention on the sophisticated tampering techniques that could appear in real splices. We believe that this is the main reason why our MAG framework achieves state-of-the-art results and outperforms other deep learning methods.

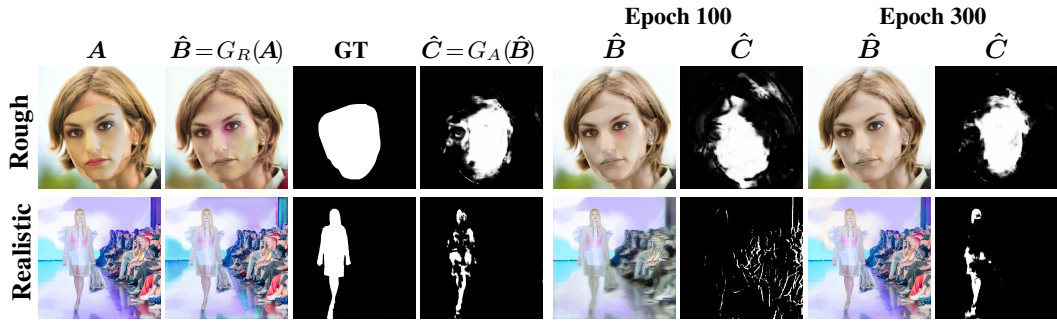

Figure 5: Performance for retoucher $G_R$ on rough and realistic splices.

Figure 6: Adaptation of annotator $G_A$ over time.

## 5 Conclusion

We showed how adversarial training based on a learning retoucher generator in the loop could help a splice localization model to learn a wide range of image manipulations. Our mixed adversarial generators extend the generative adversarial networks framework by replacing a scalar value fake prediction discriminator with a pixel-level fake region annotator. The proposed retoucher generator is trained simultaneously with an annotator generator trying to maximize the probability of the annotator to make a mistake. Such adversarial training improves the annotator splice localization rate as it observes changing image manipulation techniques through the training process. Furthermore, the competition of two generators allows the retoucher generator to achieve the state-of-the-art performance in image-to-image translation tasks. Our main observation is that semantic-guided training allows our splice localization annotator to reason explicitly about splices and their semantic consistency, and achieve and surpass the state-of-the-art methods in splice localization on several challenging datasets.

## Acknowledgments

The reported study was funded by the Russian Science Foundation (RSF) according to the research project N° 19-11-11008 and the Russian Foundation for Basic Research (RFBR) according to the research project N° 17-29-04509 . We want to thank Belgian Surrealist artist René Magritte for teaching us through his art how to find the point where fantasy meets reality.

## Footnotes

[1]http://zefirus.org/MAG

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
