[Supplementary Material · MAG_Supplementary.pdf]

# (Supplementary Material)
# The Point Where Reality Meets Fantasy: Mixed Adversarial Generators for Image Splice Detection

**Vladimir V. Kniaz, Vladimir A. Knyaz**
State Res. Institute of Aviation Systems (GosNIIAS)
125319, 7, Victorenko str., Moscow, Russia
{vl.kniaz, knyaz}@gosniias.ru

**Fabio Remondino**
FBK, Trento, Italy
remondino@fbk.eu

## 1   FantasticReality Dataset

We present examples of splices and object classes annotations in our FantasticReality in Figures 1 and 2. Class annotations are labeled with colors and spliced object is labeled with grey color.

Figure 1: Examples of annotated images in our FantasticReality dataset.

Figure 2: Examples of annotated images in our FantasticReality dataset.

The dataset is divided into two splits: 'Rough' and 'Realistic'. 'Rough' split contains 8k splices with obvious artifacts such aliasing at splice edges, light and color inconsistencies. We use the 'Rough' split to allow gradual learning of our retouching generator $G_R$. The 'Realistic' split provides 8k splices that were retouched manually to be visually indistinguishable from the authentic background image. We use the 'Realistic' split to test our model and baselines.

## 2  Modern-to-Retro Dataset

We present examples of splices and object classes annotations in the *modern→retro* in Figure 3. We manually generated ground truth annotations of instance and class labels for ten object categories: person, car, truck, van, bus, building, cat, dog, tram, boat.

## 3  Evaluation Protocol

**Datasets.** *CASIA v2.0* [1] consists of 5123 tampered images with various kinds of objects and tampering artifacts. The dataset does not provide the ground truth maps of tampered regions. We generate ground truth segmentation for CASIA v2.0 dataset similar to [2] by subtraction of tampered and authentic images. *Carvalho* [3] dataset includes 100 spliced human portraits and ground truth tampering masks. *Columbia* [4] dataset consists of 180 splices with objects of different categories. *Realistic Tampering* [5] dataset includes 165 images with challenging splices. All splices are carefully processed to be indistinguishable from the background image. For the fair evaluation, we downscale all images to match the input size $512 \times 512$ of our annotator generator $G_R$. We use the downscaled images to evaluate all baselines and our framework.

Figure 3: Examples of annotated images in our *modern→retro* dataset.

**Baselines.** We compare our model with five modern splice detection methods: `ManTra` [6],`LSC` [7], `MFCN` [8], `NOI` [9], `CFA` [10], `DCT` [11]. ManTra-Net (`ManTra`) [6] is a self-supervised model that learns to classify 385 image manipulation types. Learned Self-Consistency (`LSC`) [7] is a self-supervised model. Multi-Task Fully Convolutional Network (`MFCN`) [8] leverages a deep two-stream architecture to predict splice mask and splice edge mask. Noise Variance (`NOI`) [9] leverages wavelet analysis to detect inconsistency in noise patterns. Color Filter Array (`CFA`) [10] searches for inconsistencies in artifacts of demosaicing algorithm to detect tampered regions. JPEG `DCT` [11] leverages inconsistencies of JPEG blocking artifact to detect tampered image regions. For the `LSC` algorithm, we use a pertained model provided by authors. We implemented the `MFCN` model and train it on the training split of our FantasticReality dataset. We train our `MAG` model on the 'Rough' split of our FantasticReality dataset. We use a batch size of one an an Adam solver with initial learning rate of $2 \cdot 10^{-4}$. We trained our `MAG` model for 400 epochs.

If two images are used for splice generation, the choice of 'authentic' and 'tampered' regions is ambiguous. To avoid ambiguity, we follow the method proposed in [7]. Namely, we compare the areas of the 'background' image and the 'pasted' images. We define the smaller region as the tampered. If the regions are equal, we calculate the mAP score for the original tampering mask and an inverted mask. We use the higher score and term it permuted mAP (p-mAP) similar to [7].

## 4 Network Architecture

The architecture of our generator U-Net-UC is presented in Table 1. Our main contribution to the U-Net generator [12] architecture is in the decoder part. we replaced deconvolutional layers with an upsample layer followed by a convolutional layer, inspired by the architecture proposed in [13].

### 4.1 Semantic-guided Retoucher Generator $G_R$ Evaluation

We evaluate our retoucher generator $G_R$ ability to a perform semantic-guided image-to-image translation on a *modern→retro* task. We compare our `MAG` model against three image-to-image translation models: `CycleGAN` [14], `UNIT` [15], and `AGGAN` [16].

**Baselines.** `CycleGAN` [14] model performs unpaired image-to-image translation using cycle consistency. Unsupervised Image-to-Image Translation Networks (`UNIT`) [15] uses a shared-latent space assumption to learn a latent representation that connects images in source and target domains. Unsu-

| Name | Kernel | Str. | Ch I/O | In Res | Out Res | Input |
|------|--------|------|--------|--------|---------|-------|
| conv0 | $4 \times 4$ | 2 | 4/64 | $512 \times 512$ | $256 \times 256$ | Input image |
| conv1 | $4 \times 4$ | 2 | 4/64 | $256 \times 256$ | $128 \times 128$ | conv0 |
| conv2 | $4 \times 4$ | 2 | 64/128 | $128 \times 128$ | $64 \times 64$ | conv1 |
| conv3 | $4 \times 4$ | 2 | 128/256 | $64 \times 64$ | $32 \times 32$ | conv2 |
| conv4 | $4 \times 4$ | 2 | 256/512 | $32 \times 32$ | $16 \times 16$ | conv3 |
| conv5 | $4 \times 4$ | 2 | 512/512 | $16 \times 16$ | $8 \times 8$ | conv4 |
| conv6 | $4 \times 4$ | 2 | 512/512 | $8 \times 8$ | $4 \times 4$ | conv5 |
| conv7 | $4 \times 4$ | 2 | 512/512 | $4 \times 4$ | $2 \times 2$ | conv6 |
| conv8 | $4 \times 4$ | 2 | 512/512 | $2 \times 2$ | $1 \times 1$ | conv7 |
| upscale8 | – | – | 512/512 | $1 \times 1$ | $2 \times 2$ | conv8 |
| pad8 | – | – | 512/512 | $2 \times 2$ | $4 \times 4$ | upscale8 |
| conv_up8 | $3 \times 3$ | 1 | 512/512 | $4 \times 4$ | $2 \times 2$ | pad8 |
| upscale7 | – | – | 1024/512 | $2 \times 2$ | $4 \times 4$ | conv_up8 + conv7 |
| pad7 | – | – | 1024/512 | $4 \times 4$ | $6 \times 6$ | upscale7 |
| conv_up7 | $3 \times 3$ | 1 | 1024/512 | $6 \times 6$ | $4 \times 4$ | pad7 |
| upscale6 | – | – | 1024/512 | $4 \times 4$ | $8 \times 8$ | conv_up7 + conv6 |
| pad6 | – | – | 1024/512 | $8 \times 8$ | $10 \times 10$ | upscale6 |
| conv_up6 | $3 \times 3$ | 1 | 1024/512 | $10 \times 10$ | $8 \times 8$ | pad6 |
| upscale5 | – | – | 1024/512 | $8 \times 8$ | $16 \times 16$ | conv_up6 + conv5 |
| pad5 | – | – | 1024/512 | $16 \times 16$ | $18 \times 18$ | upscale5 |
| conv_up5 | $3 \times 3$ | 1 | 1024/512 | $18 \times 18$ | $16 \times 16$ | pad5 |
| upscale4 | – | – | 1024/256 | $16 \times 16$ | $32 \times 32$ | conv_up5 + conv4 |
| pad4 | – | – | 1024/256 | $32 \times 32$ | $34 \times 34$ | upscale4 |
| conv_up4 | $3 \times 3$ | 1 | 1024/256 | $34 \times 34$ | $32 \times 32$ | pad4 |
| upscale3 | – | – | 512/128 | $32 \times 32$ | $64 \times 64$ | conv_up4 + conv3 |
| pad3 | – | – | 512/128 | $64 \times 64$ | $66 \times 66$ | upscale3 |
| conv_up3 | $3 \times 3$ | 1 | 512/128 | $66 \times 66$ | $64 \times 64$ | pad3 |
| upscale2 | – | – | 256/64 | $64 \times 64$ | $128 \times 128$ | conv_up3 + conv2 |
| pad2 | – | – | 256/64 | $128 \times 128$ | $130 \times 130$ | upscale2 |
| conv_up2 | $3 \times 3$ | 1 | 256/64 | $130 \times 130$ | $128 \times 128$ | pad2 |
| upscale1 | – | – | 128/64 | $128 \times 128$ | $256 \times 256$ | conv_up2 + conv1 |
| pad1 | – | – | 128/64 | $256 \times 256$ | $258 \times 258$ | upscale1 |
| conv_up1 | $3 \times 3$ | 1 | 128/64 | $258 \times 258$ | $256 \times 256$ | pad1 |
| upscale0 | – | – | 128/2+K | $256 \times 256$ | $512 \times 512$ | conv_up1 + conv0 |
| pad0 | – | – | 128/2+K | $512 \times 512$ | $514 \times 514$ | upscale0 |
| conv_up0 | $3 \times 3$ | 1 | 128/2+K | $514 \times 514$ | $512 \times 512$ | pad0 |

Table 1: The U-Net-UC generator architecture.

pervised Attention-guided Image-to-Image Translation (`AGGAN`) [16] leverages unsupervised attention learning to perform translation focused only on the target object class.

We introduce a new '*modern→retro*' dataset with 2k images and class annotations for training models to translate modern cityscapes to retro photos of 1920-1930s (see supplementary material). This task is challenging as the model should translate the appearance of multiple object classes, while keeping the resulting image realistic.

**Qualitative results.** Results of translation are presented in Figure 4. While our model does not receive semantic labeling as an input, the semantic loss forces it to keep objects in the output image semantically consistent. All baselines fail to match old and new cars to perform *modern→retro* translation. Only `UNIT` model synthesizes an old car but in the wrong place. Our `MAG` model is focused on matching the semantic labels of input and target domains. Our structured loss combining adversarial and semantic losses makes the output of our model both realistic and semantically consistent.

| Input | CycleGAN [14] | UNIT [15] | AGGAN [16] | Ours |
|---|---|---|---|---|

Figure 4: Comparison against three state-of-the-art methods on various image-to-image translation tasks. Our results are shown in the last column. See supplementary material for video example.

**Quantitative results.** We use Kernel Inception Distance [17] (KID) and user perceptual realism judgment to evaluate our retoucher generator $G_R$ quantitatively. The KID represents the squared maximum mean discrepancy between deep feature representation of evaluated images. We compute the KID between the generated images and images from the target domain. To evaluate perceptual similarity, we run the test on the Amazon Mechanical Turk (AMT), similar to [18]. Quantitative results are presented in Figure 5 and Table 2. Our retouching generator $G_R$ achieves the lowest KID distance. The UNIT framework is the next best performing method.

Figure 5: Realism vs. KID for synthesized images. Lower KID and high realism is better

| Method | Realism<br>AMT Fooling Rate [%] | Distance<br>KID |
|---|---|---|
| Random real images | 50.0% | |
| AGGAN [16] | 3.44 | $18.45 \pm 0.73$ |
| CycleGAN [14] | 1.14 | $17.92 \pm 0.43$ |
| UNIT [15] | 12.32 | $12.16 \pm 0.51$ |
| Ours | **20.26** | $\mathbf{7.23 \pm 0.67}$ |

Table 2: Perceptual realism and Kernel Inception Distance $\times\ 100 \pm$ std. $\times\ 100$ for different image translation algorithms.