[Reviews · NeurIPS 2019]

Reviewer 1



The paper provides a new splice detection approach that consists of two generators trained adversarially. The retoucher tries to tamper with the input image by adding a splice while the annotator tries to detect splices (as well as the object classes in the image). Pros: + Nice and novel idea. It is also intuitively simple: train the splice detector against the various splice types learn by the retoucher. + Sound and effective architecture, cleverly using existing tricks in the literature. Clear exposition. + Extensive experimental analysis against several state of the art baselines on multiple datasets. Convincing results both in qualitative and quantitative terms. Cons: - The paper could benefit from more analysis on why it is so effective. For example, the authors could show the tampered images by G_R, performance for each type of splice (rough/realistic), how G_A adapts over time, does it overfit when G_R becomes very effective (i.e. is G_A still able to detect less sophisticated splices)? Also, the training of G_A seems broken when no G_R (it outputs the exact same mask) but the authors do not explicitly mention this. - Citation for [9] is wrong, I believe it should be Mejjati et al. "Unsupervised Attention-guided Image-to-Image Translation", NeurIPS 2018, also referred to as AGGAN later on. - Some details: losses for discriminators not explicitly stated, unconventional notation for L1 distance (should be '-', not ','), wrong GT mask for third column in Fig. 3. The author's response successfully addresses most of the reviewers' concerns. I keep my acceptance score.

Reviewer 2



A main selling point of this paper is that it significantly improves image splice detection on several standard datasets. This is achieved using clever engineering with an elaborate architecture and loss functions. I don't think there is a particularly deep novel insight in this paper, but the intuition behind the design decisions makes sense, and it seems to pay off in the evaluations. The paper is mostly well written. I think it might be difficult though to reproduce the results based on the information in the paper only due to the complexity of the system. Therefore, it would be important that code will published also. In summary, this paper makes significant progress in an important practical problem. It presents a well engineered system with a reasonable evaluation. There I think it could be published at NIPS.

Reviewer 3



The paper proposed a splice detection system that is designed based on the existing unsupervised image-to-image translation frameworks. Specifically, it assumes two domains of images: the authentic domain and the forged domain. It utilizes two generators where one generator aims at translating a forged image to a corresponding authentic image while the other generator aims at producing the semantic segmentation mask and the binary mask of the forged region if any in the input image. By teaming up with a discriminator, they serves as an adversary to the first generator. In some sense, the first generator acts as a data augmenter to help the second generator performs better in detecting forged regions. - In terms of originality, the design proposed in the paper appears to be new and makes intuitive sense. - In terms of clarity of the presentation, it is below the publication standard. Vague descriptions are abundant in the manuscript. The notation is also hard to follow. For example, Equation (2) and (3) are difficult to understand. It is unclear how the label loss is computed. - In terms of significant, the baselines the paper compares to for the forged region detection task, which is the main task considered in the paper, appear to be simple methods based on local statistics analysis. It is not surprised that a deep learning based method performs better. The paper should consider stronger baseline that are based on deep learning. It is unclear why the paper includes a comparison to unsupervised image-to-image translation model. Achieving better performance on the image translation task is not helping the paper as 1) this task is just digression from the main topic, and 2) the achieved better performance is largely due to the use of segmentation mask in the training. It is expected. It is not supporting the proposed paper.

[Author Response · NeurIPS 2019]

| Dataset | CASIA v2.0 | | | Columbia | | | RT | | | Carvalho | | | FantasticReality | | |
|---|---|---|---|---|---|---|---|---|---|---|---|---|---|---|---|
| | mAP | p-mAP | cIOU | mAP | p-mAP | cIOU | mAP | p-mAP | cIOU | mAP | p-mAP | cIOU | mAP | p-mAP | cIOU |
| ManTra | 0.40 | 0.40 | 0.45 | 0.48 | 0.48 | 0.58 | **0.50** | 0.50 | 0.54 | 0.33 | 0.33 | 0.38 | 0.57 | 0.57 | 0.73 |
| Ours | **0.74** | **0.74** | **0.76** | **0.69** | **0.69** | **0.77** | 0.50 | **0.51** | **0.55** | **0.48** | **0.48** | **0.56** | **0.61** | **0.61** | **0.76** |

Table 1: **Splice Localization:** MAG vs. `ManTra-Net` [1] following the protocol defined in supplementary material.

Dear reviewers, we very much appreciate your valuable comments, time, and effort. Below we provide a detailed
response to each reviewer.

**R6:** In terms of significant, the baselines the paper compares to for the forged region detection task, which is the main
task considered in the paper, appear to be simple methods based on local statistics analysis.

We compare our MAG framework to two modern state-of-the-art deep learning splice detection frameworks: `LSC` is
based on a Siamese network and was presented at ECCV 18, `MFCN` is a Multi-Task Fully Convolutional Network that
was published in 2018 ([1,2] in our paper). We provide a comparison to non-deep learning methods to be consistent with
the `LSC` paper. We compared our MAG framework to new deep learning-based `ManTra-Net` [1] that was not published
during the submission and will be presented at CVPR 19 (see Table 1). Our method outperforms three state-of-the-art
deep learning frameworks in terms of mAP, permuted mAP, and per class Intersection over Union (cIOU).

**R6:** Vague descriptions are abundant in the manuscript. The notation is also hard to follow. For example, Equation (2)
and (3) are difficult to understand. It is unclear how the label loss is computed.

We are sorry for typos in equation (2) and (3), there should be '-', not ',' as it is noted by Reviewer 4. Equation (2) and
(3) define a classical L1 distance. We will appreciate specific comments on the vague descriptions in the paper. We will
clarify them in the camera-ready version. The class label loss is given by equation (6). The specific predicted class
labels are given by $\hat{C}_i$ for $i \in \{3, 4, \ldots, 2 + K\}$, where $K$ is the number of classes ($K = 10$ in our experiments).

**R5 / R6 / R4:** The paper could benefit from more analysis on why it is so effective. I think the image-to-image
translation part does not add much to the paper.

We can remove the image-to-image translation part and provide an extended analysis of the of (1) failure cases and
(2) why is our framework is so effective. Examples in Figures 1 and 2 demonstrates how our retoucher $G_R$ gradually
removes the tampering artifacts in the input splice $A$ with an increasing epoch. While other deep learning splice
detection methods receive both realistic and rough splices from the first training epoch, our annotator $G_A$ sees only
rough splices at the first epoch. With an increasing epoch, retoucher $G_R$ produces more complicated splices, which
allows $G_A$ to focus attention on the sophisticated tampering techniques that could appear in real splices. We believe
that this is the main reason why our MAG framework achieves the state-of-the-art results and outperforms other deep
learning methods.

Figure 1: Performance for retoucher $G_R$ on rough and realistic splices.

Figure 2: Adaptation of annotator $G_A$ over time.

# References

[1] Wael AbdAlmageed Yue Wu and Premkumar Natarajan. Mantra-net: Manipulation tracing network for detection
and localization of image forgerieswith anomalous features. In *The IEEE Conference on Computer Vision and*
*Pattern Recognition (CVPR)*, 2019.


[Meta-Review · NeurIPS 2019]

This work addresses the task of detecting the tampering of images in which multiple images are combined into one (image splicing). To this end the authors introduce a novel generative adversarial training technique consiting of four models that are trained simultaneously. The method is shown to perform very well on existing data sets and a novel large scale image manipulation dataset. The experiments are detailed and code and data set will be made available. Three expert reviewers initially assessed the work as 8/7/5, and the authors provided a detailed rebuttal; two reviewers acknowledge reading the rebuttal and one adjusted the score, for a final rating of 8/7/6. Remaining concerns are around clarity of writing and limited analysis studying the high effectiveness of the method in more detail. Overall, this is an important problem and the paper makes a clear contribution, both in terms of an interesting novel method and a new data set.